# Adaptive Graph Rewiring to Mitigate Over-Squashing in Mesh-Based GNNs for Fluid Dynamics Simulations

## Abstract

Mesh-based simulation using Graph Neural Networks (GNNs) has been recognized as a promising approach for modeling fluid dynamics. However, the mesh refinement techniques which allocate finer resolution to regions with steep gradients can induce the over-squashing problem in mesh-based GNNs, which prevents the capture of long-range physical interactions. Conventional graph rewiring methods attempt to alleviate this issue by adding new edges, but they typically complete all rewiring operations before applying them to the GNN. These approaches are physically unrealistic, as they assume instantaneous interactions between distant nodes and disregard the distance information between particles. To address these limitations, we propose a novel framework, called Adaptive Graph Rewiring in Mesh-Based Graph Neural Networks (AdaMeshNet), that introduces an adaptive rewiring process into the message-passing procedure to model the gradual propagation of physical interactions. Our method computes a rewiring delay score for bottleneck nodes in the mesh graph, based on the shortest-path distance and the velocity difference. Using this score, it dynamically selects the message-passing layer at which new edges are rewired, which can lead to adaptive rewiring in a mesh graph. Extensive experiments on mesh-based fluid simulations demonstrate that AdaMeshNet outperforms conventional rewiring methods, effectively modeling the sequential nature of physical interactions and enabling more accurate predictions. Our source code is available at `https://anonymous.4open.science/r/AdaMeshNet-9321`.

## 1 Introduction

Fluid dynamics has seen various attempts to solve the Navier-Stokes equations (Temam, 1977). Since analytical solutions for complex physics are unobtainable, numerical methods such as the finite element method (FEMs) (Madenci & Guven, 2006; Stolarski et al., 2018; Dhatt et al., 2012) have been widely adopted to solve the differential equations by discretizing them in space and time. As a key strategy for enhancing the accuracy of these numerical methods, the mesh refinement technique (Löhner, 1995; Liu et al., 2022) generates adaptive meshes by increasing the resolution of specific regions that require detailed analysis, such as areas with sharp gradients involving unstructured surfaces in complex dynamics problems. The adaptive meshes are used to focus computational resources on the most critical areas, which enables high accuracy in mesh simulations without the need to compute the entire domain at high resolution, even with limited computational power.

Recently, graph neural networks (GNNs) have been widely used for mesh simulations by leveraging these advantages of adaptive meshes. In particular, MeshGraphNets (MGN) (Pfaff et al., 2020) have proven effective at approximating simulation results on unstructured meshes by propagating local physical interactions between nodes via message passing (Sanchez-Gonzalez et al., 2020; Fortunato et al., 2022; Nabian et al., 2024).

A key challenge in applying GNNs to fluid dynamics simulations is balancing mesh refinement with the propagation of interactions. Specifically, regions with sharp gradients, such as boundary layers and turbulence, require higher-density mesh structures for accurate simulation (Katz & Sankaran, 2011; Baker, 2005). However, these fine mesh structures cause the **over-squashing** problem when physical interactions are propagated through the graph (Topping et al., 2022; Di Giovanni et al., 2023; Black et al., 2023). Over-squashing refers to the progressive compression of information from

distant nodes as it passes through multiple layers in GNNs. This compression becomes more severe in the fine mesh areas, which makes it difficult to capture long-range interactions. In particular, fluid dynamics simulations require mesh refinement techniques to accurately capture complex flow phenomena, which makes this challenge especially pronounced compared to many other domains.

To solve the over-squashing problem, several graph rewiring approaches that account for graph topology have been developed (Gasteiger et al., 2019; Karhadkar et al., 2023; Nguyen et al., 2023). Recently, PIORF (Yu et al., 2025) introduced a graph rewiring approach specifically designed for fluid dynamics simulations, which considers not only graph topology but also physical quantities of the fluid. However, in existing approaches, all rewiring occurs before GNN training for fluid simulations and it forces nodes to interact as if they were immediate neighbors. This leads the model to lose information about their actual physical distance and gradual propagation, which is unrealistic for long-range interactions in fluids. In reality, phenomena such as boundary layers and turbulence affect distant particles only after a certain delay, since their influence propagates gradually through sequential collisions among neighboring particles. This highlights the need for a new rewiring approach that explicitly accounts for the gradual propagation of physical interactions without loss of inter-node distance information during long-range interactions.

In this work, we theoretically demonstrate the over-squashing phenomenon inherent in MGN, which is widely used as a mesh-based GNN model. Additionally, to address this issue, we propose a novel framework, called Adaptive Graph Rewiring to Mitigate Over-Squashing in Mesh-Based GNNs for Fluid Dynamics Simulations (AdaMeshNet). The key idea is to dynamically rewire new edges during the message-passing process by considering the gradual propagation of physical interactions in fluid simulations. We first detect bottleneck nodes in the graph based on Ollivier–Ricci curvature (ORC) (Ollivier, 2009). We then compute the distances between these bottleneck nodes and nodes with large velocity differences, and subsequently calculate the rewiring delay score using both the distances and the velocity differences. The rewiring delay score quantifies the degree of rewiring delay and serves to determine the layer at which rewiring should be performed during the message-passing process. Based on the computed rewiring delay scores, we rewire bottleneck nodes with nodes of high velocity gradients at each layer of the message passing process. This approach applies rewiring delays based on curvature and physical quantity, which enables simulations to consider the gradual propagation of interactions. Therefore, our model provides an effective solution to the over-squashing problem in fluid simulations by performing adaptive graph rewiring in the message passing process.

To validate our approach, we conducted extensive experiments on two fluid dynamics datasets: CylinderFlow and Airfoil. Our AdaMeshNet framework was compared with leading static rewiring methods, all implemented on the MGN model. The results demonstrate that AdaMeshNet achieves more accurate predictions of key physical quantities, such as velocity and pressure. Furthermore, it produces velocity contours that more closely match the ground truth, particularly in capturing complex phenomena like wavelike propagation. These findings highlight our model's ability to effectively solve the over-squashing problem by considering the gradual nature of physical interactions, a crucial aspect often overlooked by existing methods.

In summary, our main contributions are summarized as follows:

- We provide a theoretical demonstration of the over-squashing problem in MGN.
- We propose an adaptive rewiring method that considers the gradual propagation of physical interations to address the over-squashing problem in fluid simulations.
- We demonstrate that our model outperforms existing rewiring methods in our experiments.

## 2 PRELIMINARIES

### 2.1 PROBLEM DEFINITION

The goal of our task is to train a model that predicts the dynamic quantity of the mesh at time $t + 1$, using the current mesh $\mathcal{M}_t$ at time $t$ and past meshes $\{\mathcal{M}_{t-1}, \mathcal{M}_{t-2}, \ldots, \mathcal{M}_{t-h}\}$. Our fluid dynamics simulations are based on the Euler system, which models physical quantities that change over time on the fixed mesh coordinates and incorporates these changes into the simulation.

The mesh $\mathcal{M}_t$ is transformed into a multi-graph $\mathcal{G} = (\mathcal{V}, \mathcal{E}, A)$. The mesh nodes and edges are mapped to graph nodes $\mathcal{V}$ and bidirectional edges $\mathcal{E}$, respectively. $A$ denotes the adjacency matrix,

and we define $\tilde{A} = A + I$, which is the adjacency matrix augmented with self-loops. We then define $\hat{A}$ as the normalized augmented adjacency matrix, i.e., $\hat{A} = \tilde{D}^{-\frac{1}{2}} \tilde{A} \tilde{D}^{-\frac{1}{2}}$, where $\tilde{D} = D + I$ and $D$ is the diagonal degree matrix. Each node has node features consisting of the dynamic feature $q_i$ and a one-hot vector that represents the node type $n_i$, which includes fluid, wall, inflow, and outflow nodes. Each edge has features $m_{ij}$, which include connection information such as the distance between two particles, as well as the relative displacement vector $\mathbf{d}_{ij} = \mathbf{d}_i - \mathbf{d}_j$ and its norm $|\mathbf{d}_{ij}|$ to achieve spatial invariance.

## 2.2 MESHGRAPHNETS

MeshGraphNets (MGN) (Pfaff et al., 2020) is a GNN model designed to predict the dynamics of physical systems based on mesh simulations. The model first encodes the physical simulation data as graphs. Then, it updates node and edge embeddings through multi-layer message passing and predicts the physical quantities at the next time step based on the embeddings.

The processor, which plays a central role in this message-passing mechanism, is composed of $L$ message-passing blocks. Each block sequentially performs edge and node updates to propagate information through the graph. Specifically, the edge embedding at layer $l + 1$ is updated through $f_E$, which takes as input two node embeddings at layer $l$ and the edge embedding connecting them. Next, the node embedding at layer $l + 1$ is updated through $f_V$, which takes as input the node embedding at layer $l$ and the updated edge embedding at layer $l + 1$. The detailed procedure of the processor is as follows:

$$\mathbf{e}_{ij}^{(\ell+1)} = f_E \left( \mathbf{e}_{ij}^{(\ell)}, \mathbf{h}_i^{(\ell)}, \mathbf{h}_j^{(\ell)} \right), \quad \mathbf{h}_i^{(\ell+1)} = f_V \left( \mathbf{h}_i^{(\ell)}, \sum_{j=1}^{n} \hat{A}_{ij} \, \mathbf{e}_{ij}^{(\ell+1)} \right), \tag{1}$$

where $\mathbf{h}_i^{(l)}$ and $\mathbf{h}_j^{(l)}$ denote the node embeddings at layer $l$, and $\mathbf{e}_{ij}^{(l)}$ denotes the edge embedding at layer $l$. $f_E$ and $f_V$ are implemented as multi-layer perceptrons (MLPs) with residual connections.

## 2.3 OLLIVIER–RICCI CURVATURE ON GRAPHS

The Ricci curvature in differential geometry represents the dispersion of geodesics on a Riemannian manifold. Ollivier-Ricci curvature (ORC) (Ollivier, 2009) extends Ricci curvature to graphs by replacing geodesics with shortest paths between nodes, and by interpreting dispersion in terms of the probability distribution of a random walk. Given a graph $\mathcal{G} = (\mathcal{V}, \mathcal{E})$ and nodes $i, j \in \mathcal{V}$, the Ollivier-Ricci curvature (ORC) $\kappa(i, j)$ of an edge $(i, j) \in \mathcal{E}$ is computed as follows:

$$\kappa(i, j) = 1 - \frac{W_1(P_i, P_j)}{d_{\mathcal{G}}(i, j)}, \tag{2}$$

where $W_1$ is the 1st-order Wasserstein distance, $P_i$ denotes the probability distribution of a random walk starting from node $i$, and $d_{\mathcal{G}}(i, j)$ is the shortest path distance between nodes $i$ and $j$. The 1st-order Wasserstein distance $W_1(P_i, P_j)$ between $P_i$ and $P_j$ is computed as follows:

$$W_1(P_i, P_j) = \inf_{\pi \in \Pi(P_i, P_j)} \left( \sum_{(p,q) \in \mathcal{V}^2} \pi(p, q) d_{\mathcal{G}}(p, q) \right), \tag{3}$$

where $\Pi(P_i, P_j)$ is the set of joint probability distributions that have $P_i$ and $P_j$ as their marginals. The probability $P_i(p)$ that a 1-step random walker starting from node $i$ reaches node $p$ is defined as follows:

$$P_i(p) = \begin{cases} \frac{1}{\deg(i)} & \text{if } p \in \mathcal{N}_i \\ 0 & \text{if } p \notin \mathcal{N}_i, \end{cases} \tag{4}$$

where $\deg(i)$ denotes the degree of node $i$, and $\mathcal{N}_i$ represents the set of neighboring nodes of $i$. Equation 4 indicates that the random walker moves to one of the neighboring nodes with equal probability.

The ORC $\kappa(i, j)$ represents the degree of dispersion of geodesics, with different ranges indicating distinct structural implications of information flow. Its value indicates whether information is likely to converge, flow stably, or diverge as follows: **1)** $\kappa(i, j) > 0$ (Convergence): When the ORC value

is positive, geodesics tend to converge. This suggests that information is likely to concentrate at certain points, which can lead to efficient integration and processing. **2)** $\kappa(i,j) = 0$ (Stable Flow): A zero value indicates that geodesics remain parallel. This implies a stable and uniform flow of information without the formation of bottlenecks. **3)** $\kappa(i,j) < 0$ (Divergence): When the ORC value is negative, geodesics tend to diverge. This suggests the presence of bottlenecks or structural constraints that can reduce the efficiency of information transfer. It is also worth noting that Topping et al. (2022) observed that highly negative ORC values can contribute to the over-squashing problem, a phenomenon where information becomes compressed and difficult to propagate effectively.

## 3 METHODOLOGY

### 3.1 ANALYSIS OF THE OVER-SQUASHING PHENOMENON IN MESH-BASED GNN

We provide a theoretical analysis of how well MGN captures long-range interactions in scenarios with a large number of distant neighbor nodes. We assume that the graph $\mathcal{G}$ has node features $X \in \mathbb{R}^{n \times p_0}$, where $\mathbf{x}_i \in \mathbb{R}^{p_0}$ is the feature vector of node $i = 1, \ldots, n = |\mathcal{V}|$. The hidden representations $\mathbf{h}_i^{(\ell)}$ and $\mathbf{e}_{ij}^{(\ell)}$, as computed by Equation 1, are differentiable with respect to the input node features $\{\mathbf{x}_1, \ldots, \mathbf{x}_n\}$, provided that $f_V$ and $f_E$ are differentiable functions. We evaluate how much the node $\mathbf{h}_i^{(\ell)}$ and edge $\mathbf{e}_{ij}^{(\ell)}$ are influenced by the input features $\mathbf{x}_s$ of a node $s$ located at distance $r$ from node $i$. To this end, we utilize the Jacobians $\partial \mathbf{h}_i^{(r)} / \partial \mathbf{x}_s$ and $\partial \mathbf{e}_{ij}^{(r)} / \partial \mathbf{x}_s$ as follows.

**Lemma 1.** *Assume a message-passing scheme for mesh simulation in Equation 1. Let $i, j, s \in \mathcal{V}$ be nodes in the graph $\mathcal{G}$, where $j$ is a neighbor of $i$ and the $s$ is an $r$-hop neighbor of $i$, i.e., $j \in \mathcal{N}_i$ and $d_{\mathcal{G}}(i,s) = r$. If $|\partial_2 f_V| \leq \alpha_e$, $|\partial_3 f_E| \leq \beta_h$ for $0 \leq l \leq r - 1$, then*

$$\left| \frac{\partial \mathbf{h}_i^{(r)}}{\partial \mathbf{x}_s} \right| \leq (\alpha_e \beta_h)^r \left( \hat{A}^r \right)_{is}, \quad \left| \frac{\partial \mathbf{e}_{ij}^{(r)}}{\partial \mathbf{x}_s} \right| \leq \alpha_e^{r-1} \beta_h^r \left( \hat{A}^{r-1} \right)_{js}.$$

*Proof.* Since $d_{\mathcal{G}}(i,s) = r$, note that the Jacobians $\frac{\partial \mathbf{h}_i^{(r-1)}}{\partial \mathbf{x}_s}$, $\frac{\partial \mathbf{h}_j^{(r-2)}}{\partial \mathbf{x}_s}$ and $\frac{\partial \mathbf{e}_{ij}^{(r-1)}}{\partial \mathbf{x}_s}$ are zero matrices. Then, we can recursively expand $\frac{\partial \mathbf{e}_{ij}^{(r)}}{\partial \mathbf{x}_s}$ as follows:

$$\frac{\partial \mathbf{e}_{ij}^{(r)}}{\partial \mathbf{x}_s} = \frac{\partial f_E}{\partial \mathbf{e}_{ij}^{(r-1)}} \frac{\partial \mathbf{e}_{ij}^{(r-1)}}{\partial \mathbf{x}_s} + \frac{\partial f_E}{\partial \mathbf{h}_i^{(r-1)}} \frac{\partial \mathbf{h}_i^{(r-1)}}{\partial \mathbf{x}_s} + \frac{\partial f_E}{\partial \mathbf{h}_j^{(r-1)}} \frac{\partial \mathbf{h}_j^{(r-1)}}{\partial \mathbf{x}_s}$$

$$= \frac{\partial f_E}{\partial \mathbf{e}_{ij}^{(r-1)}} \frac{\partial \mathbf{e}_{ij}^{(r-1)}}{\partial \mathbf{x}_s} + \frac{\partial f_E}{\partial \mathbf{h}_i^{(r-1)}} \frac{\partial \mathbf{h}_i^{(r-1)}}{\partial \mathbf{x}_s} + \frac{\partial f_E}{\partial \mathbf{h}_j^{(r-1)}} \left( \frac{\partial f_V}{\partial \mathbf{h}_j^{(r-2)}} \frac{\partial \mathbf{h}_j^{(r-2)}}{\partial \mathbf{x}_s} + \frac{\partial f_V}{\partial \mathbf{z}_j^{(r-1)}} \sum_k \hat{A}_{jk} \frac{\partial \mathbf{e}_{jk}^{(r-1)}}{\partial \mathbf{x}_s} \right)$$

$$= \frac{\partial f_E}{\partial \mathbf{h}_j^{(r-1)}} \frac{\partial f_V}{\partial \mathbf{z}_j^{(r-1)}} \sum_k \hat{A}_{jk} \frac{\partial \mathbf{e}_{jk}^{(r-1)}}{\partial \mathbf{x}_s} = \cdots = \sum_{j_2, \ldots, j_{r-1}} \hat{A}_{jj_2} \hat{A}_{j_2 j_3} \cdots \hat{A}_{j_{r-1}s} \cdot J_{ij_2 \cdots j_{r-1}s}(X),$$

where $\mathbf{z}_j^{(r-1)} = \sum_k \hat{A}_{jk} \mathbf{e}_{jk}^{(r-1)}$ and $J_{jj_2 \cdots j_{r-1}s}(X)$ is the product of $r$ third partial derivatives of $f_E$ and $r - 1$ second partial derivatives of $f_V$ with $j_l$ indicating the index of $i$'s $l$-hop neighbors.

Since $|J_{jj_2 \cdots j_{r-1}s}(X)| \leq \alpha_e^{r-1} \beta_h^r$ holds, we obtain

$$\left| \frac{\partial \mathbf{e}_{ij}^{(r)}}{\partial \mathbf{x}_s} \right| \leq \sum_{j_2, \ldots, j_{r-1}} \hat{A}_{jj_2} \hat{A}_{j_2 j_3} \cdots \hat{A}_{j_{r-1}s} \alpha_e^{r-1} \beta_h^r = \alpha_e^{r-1} \beta_h^r \left( \hat{A}^{r-1} \right)_{js}.$$

Using this result, we can also derive the upper bound of $\left| \frac{\partial \mathbf{h}_i^{(r)}}{\partial \mathbf{x}_s} \right|$ as follows:

$$\left| \frac{\partial \mathbf{h}_i^{(r)}}{\partial \mathbf{x}_s} \right| = \left| \frac{\partial f_V}{\partial \mathbf{h}_i^{(r-1)}} \frac{\partial \mathbf{h}_i^{(r-1)}}{\partial \mathbf{x}_s} + \frac{\partial f_V}{\partial \mathbf{z}_i^{(r)}} \sum_j \hat{A}_{ij} \frac{\partial \mathbf{e}_{ij}^{(r)}}{\partial \mathbf{x}_s} \right|$$

$$\leq \alpha_e \sum_{j, j_2, \ldots, j_{r-1}} \hat{A}_{ij} \hat{A}_{jj_2} \cdots \hat{A}_{j_{r-1}s} \alpha_e^{r-1} \beta_h^r = (\alpha_e \beta_h)^r \left( \hat{A}^r \right)_{is}.$$

$\square$

Lemma 1 shows that if $f_V$ and $f_E$ have bounded derivatives, the extent of message propagation in a mesh-based GNN is controlled by powers of $\hat{A}$. Intuitively, as the hop distance $r$ increases, the number of $r$-hop neighbors within the receptive field $B_r(i) = \{j \in \mathcal{V} \mid d_{\mathcal{G}}(i, j) \leq r\}$ grows rapidly. Because information from this expanding set of neighbors must ultimately be compressed into a fixed-size vector, the influence of each individual neighbor necessarily diminishes with increasing $r$. This diminishing influence is reflected in the upper bound of the Jacobian terms derived in Lemma 1, which decay exponentially as a function of the hop distance. This result is precisely what gives rise to the over-squashing phenomenon. More detailed derivation is provided in Appendix A.

## 3.2 Adaptive Graph Rewiring in Mesh-Based GNN

To address the over-squashing problem analyzed in Section 3.1, we propose a novel graph rewiring method for mesh simulations. Recall that in existing rewiring methods (Gasteiger et al., 2019; Karhadkar et al., 2023; Nguyen et al., 2023; Yu et al., 2025), all rewiring occurs before GNN training. This causes two distant particles to interact instantaneously, as if they were neighboring particles, which does not sufficiently reflect actual physical conditions. For example, boundary layers or turbulence that can affect distant particles propagate their influence sequentially through collisions between adjacent particles, leading to a certain delay before the influence reaches the distant particles. Therefore, we propose adaptive graph rewiring that considers the gradual propagation of physical interations in mesh-based GNN. Figure 1 illustrates the differences between existing rewiring methods and our proposed approach within a mesh graph.

### 3.2.1 Preprocessing

**Identifying bottleneck nodes.** We identify bottleneck nodes that cause over-squashing in the mesh graph based on ORC. We calculate obtain the node-level curvature $\gamma_i$ to summarize the local geometry around node $i$ as follows:

$$\gamma_i = \frac{1}{|\mathcal{N}_i|} \sum_{j \in \mathcal{N}_i} \kappa(i, j), \tag{5}$$

which represents the mean of the $\kappa$ values of all edges connected to node $i$. Based on the computed $\gamma_i$, we define the set of nodes $\mathcal{V}_{\text{lowORC}} \subset \mathcal{V}$ whose curvatures belong to the lowest $a\%$ as follows:

$$\mathcal{V}_{\text{lowORC}} = \{v_i \in \mathcal{V} \mid \gamma_i \leq \text{Percentile}_a(\{\gamma_j\}_{j \in \mathcal{V}})\}. \tag{6}$$

**Calculating the rewiring delay score.** To avoid performing all rewiring at once, we dynamically rewire each edge during the message-passing process. Therefore, we aim to compute the rewiring delay score, which indicates the degree of delay required to rewire each new edge. To this end, we first select the optimal connection pair for each bottleneck node to resolve the bottleneck. Yu et al. (2025) has demonstrated that rewiring nodes with large velocity differences can effectively resolve the over-squashing problem in fluid simulations. Inspired by this, we determine the optimal connection node $v_{i^*}$ based on the velocity difference for each $v_i \in \mathcal{V}_{\text{lowORC}}$ as follows:

$$i^* = \underset{j \text{ s.t. } v_j \in \mathcal{V} \setminus \{v_i\}}{\text{argmax}} \|\mathbf{v}_i - \mathbf{v}_j\| \quad \forall v_i \in \mathcal{V}_{\text{lowORC}}, \tag{7}$$

where $\mathbf{v}_i$ and $\mathbf{v}_j$ represent the velocities of $v_i$ and $v_j$, respectively. Finally, we compute the rewiring delay score $s_{\text{delay}}(i, i^*)$ based on the velocity difference between $v_i$ and $v_{i^*}$, and the shortest path distance $d_{\mathcal{G}}$ as follows:

$$s_{\text{delay}}(i, i^*) = \min\left(\frac{\beta \cdot d_{\mathcal{G}}(i, i^*)}{\|\mathbf{v}_i - \mathbf{v}_{i^*}\|}, \ L\right), \tag{8}$$

where $L$ represents the total number of message-passing blocks and $\beta$ is a hyperparameter that controls the influence of distance and velocity. $s_{\text{delay}}(i, i^*)$ represents the degree of delay required for $v_i$ and $v_{i^*}$ to be rewired, and it determines the layer index at which the two nodes are rewired during the message-passing process. As the distance increases, $s_{\text{delay}}$ increases, and conversely, as the velocity difference increases, $s_{\text{delay}}$ decreases. Specifically, as distance increases, long-range interactions require more time to propagate, so we set $s_{\text{delay}}$ to a larger value. Additionally, as the velocity difference increases, the bottleneck node have a greater influence on long-range interactions, so we set $s_{\text{delay}}$ to a smaller value.

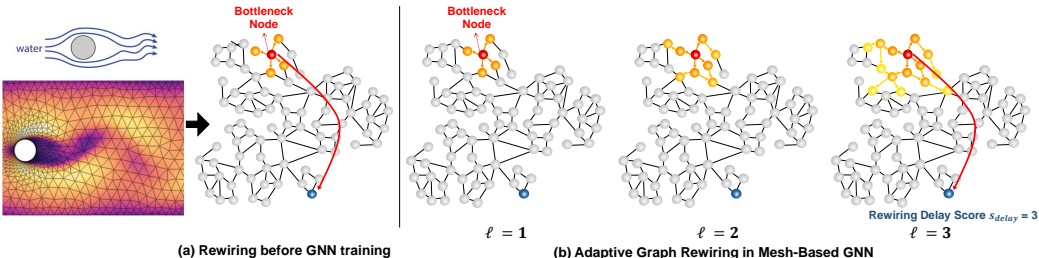

Figure 1: Comparison of static rewiring and adaptive graph rewiring (AdaMeshNet).

### 3.2.2 ENCODER

The encoder maps the node $v_i$ and edge $e_{ij}$ into latent vectors using a Multi-Layer Perceptron (MLP). Specifically, the node and edge embeddings are denoted as $\mathbf{h}_i$ and $\mathbf{e}_{ij}$, respectively, and are obtained via separate MLPs as follows:

$$\mathbf{h}_i = \text{MLP}_v(v_i), \quad \mathbf{e}_{ij} = \text{MLP}_e(e_{ij}). \tag{9}$$

### 3.2.3 PROCESSOR

**Updating nodes for rewiring.** We update the neighboring nodes based on the rewiring delay score $s_{\text{delay}}$ computed in Section 3.2.1. This update process is performed at each layer, and the overall update procedure is as follows:

$$\mathcal{N}_i^0 = \{j | (i, j) \in \mathcal{E}\}, \quad \mathcal{N}_i^{l+1} = \mathcal{N}_i^l \cup \{i^* \mid l < s_{\text{delay}}(i, i^*) \leq l+1\}. \tag{10}$$

Specifically, the initial neighboring nodes are the same as the neighbors connected by the edges $\mathcal{E}$ derived from the mesh graph. As layer $l$ increases, new neighboring nodes are added to the neighbor set $\mathcal{N}_i^l$ based on the $s_{\text{delay}}$, updating $\mathcal{N}_i^l$. As a result, a neighbor set $\mathcal{N}_i^l$ for $v_i$ is assigned at each layer $l$, and as $l$ increases, the neighboring nodes within $\mathcal{N}_i^l$ are progressively accumulated.

**Edge update.** Each message-passing block consists of an edge update and a node update. Each block contains a separate set of network parameters and is applied sequentially to the output of the previous block. The edge embedding $\mathbf{e}_{ij}^{l+1}$ at layer $l+1$ is updated based on $\mathbf{e}_{ij}^l$, $\mathbf{h}_i^l$, and $\mathbf{h}_j^l$ as:

$$\mathbf{e}_{ij}^{l+1} = f_E(\mathbf{e}_{ij}^l, \mathbf{h}_i^l, \mathbf{h}_j^l), \quad j \in \mathcal{N}_i^{l+1}. \tag{11}$$

Note that $j$ used in the edge update belongs to the neighbor set $\mathcal{N}_i^{l+1}$ of $v_i$, which is determined based on the rewiring delay score $s_{\text{delay}}$. Thus, the edge embedding $\mathbf{e}_{ij}^{l+1}$ is updated using the newly rewired neighboring nodes at each layer.

**Node update.** Next, the node embedding $\mathbf{h}_i^{l+1}$ at layer $l+1$ is updated based on $\mathbf{h}_i^l$ and $\mathbf{e}_{ij}^{l+1}$ as:

$$\mathbf{h}_i^{l+1} = f_V \left( \mathbf{h}_i^l, \sum_{j \in \mathcal{N}_i^{l+1}} \mathbf{e}_{ij}^{l+1} \right), \tag{12}$$

where $\mathbf{e}_{ij}^{l+1}$ is the edge embedding obtained from the edge update. Note that $j$ in $\mathbf{e}_{ij}^{l+1}$ belongs to $\mathcal{N}_i^{l+1}$, which is determined based on $s_{\text{delay}}$. Thus, the node embedding $\mathbf{h}_i^{l+1}$ is updated using the newly rewired edges at each layer.

### 3.2.4 DECODER AND STATE UPDATER

To predict the state at time $t+1$ from time $t$, the decoder uses an MLP to transform the outputs $o_i$, such as the velocity gradient $\hat{\mathbf{v}}_i$, density gradient $\hat{\rho}_i$, and pressure gradient $\hat{p}_i$. The updator computes the dynamic quantity $\hat{q}_i^{t+1}$ at the next step based on the outputs $o_i$ obtained from the decoder, using the forward-Euler integrator. For example, the velocity $\hat{\mathbf{v}}_i^t$ is used to compute the velocity $\hat{\mathbf{v}}_i^{t+1}$ at time $t+1$ as follows:

$$\hat{\mathbf{v}}_i^{t+1} = \hat{\mathbf{v}}_i^t + \mathbf{v}_i^t. \tag{13}$$

Finally, the output nodes $\mathcal{V}$ are updated using $q_i^{t+1}$, and $\mathcal{M}_{t+1}$ is generated based on the updated output nodes $\mathcal{V}$.

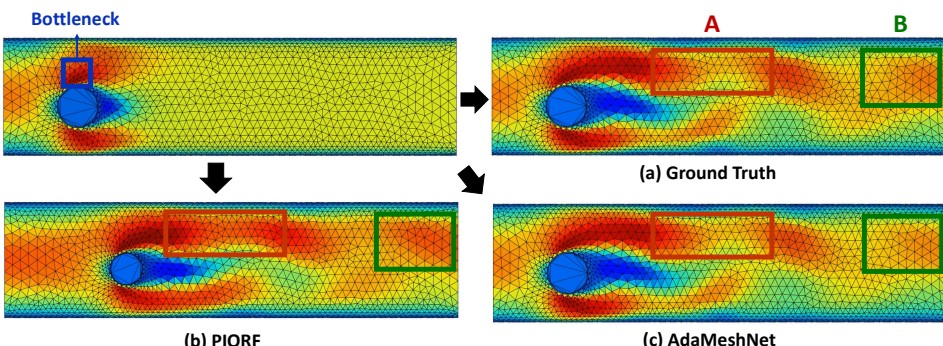

Figure 2: Physical interpretation based on visualization in Cylinder Flow.

### 3.2.5 PHYSICAL INTERPRETATION BASED ON VISUALIZATION

One of the most effective methods for analyzing fluid motion is to visualize velocity contours from fluid simulations. In Figure 2, we visualize how velocity contours propagate from the initial state in a Cylinder Flow. We compare our proposed model, AdaMeshNet, with a state-of-the-art static rewiring method, PIORF (Yu et al., 2025). In the visualizations, a red mesh indicates high velocity values, while a blue mesh indicates low velocity values. In Figure 2(b), the PIORF method fails to accurately capture the wavelike propagation of velocity in Region A. In Region B, PIORF generates an overshooting phenomenon producing velocities faster than the ground truth, since it instantly transmits interactions as if they were from adjacent particles, without considering the inherent delay. In contrast, Figure 2(c) shows that AdaMeshNet produces velocity values that are very similar to the ground truth by mimicking the physical reality of the gradual propagation of long-range interactions. Ultimately, AdaMeshNet models key fluid dynamics principles of physical interaction delay and propagation, going beyond simple graph structure improvements to enable predictions that are closer to real-world simulations.

## 4 EXPERIMENTS

### 4.1 EXPERIMENTAL SETUPS

**Datasets.** For evaluation of the models, we use Cylinderflow and Airfoil. They all operate on the basis of the Navier–Stokes equations (Temam, 1977), but the fluid behaves differently in each case. Specifically, CylinderFlow exhibits a laminar flow, where fluid particles move in a regular and orderly manner, whereas Airfoil produces a high-speed turbulent model, where fluid particles move in a disordered manner. Each dataset includes 1,000 flow results, each with 600 time steps. Details on datasets can be found in Appendix C.

**Baselines.** We use DIGL (Gasteiger et al., 2019), SDRF (Topping et al., 2022), BORF (Nguyen et al., 2023), and PIORF (Yu et al., 2025) as baselines. All of these baselines follow a static rewiring approach, completing all rewiring before applying the GNN. In our experiments, these methods were implemented based on the MGN model (Pfaff et al., 2020) as the backbone. For all models, we used 15 message-passing layers and set the hidden vector size of MLPs to 128. Details on baselines can be found in Appendix B.

### 4.2 PREDICTION OF PHYSICAL QUANTITIES

Tables 1 and 2 show the results of physical quantity predictions for the Cylinder Flow and Airfoil datasets, respectively. We measured the root-mean-square error (RMSE) for velocity, pressure, and density across a single prediction step, a 50-step rollout, and the full trajectory rollout. AdaMeshNet achieved the lowest RMSE across all metrics when compared to existing static graph rewiring methods. The superior performance of AdaMeshNet on both datasets indicates its effectiveness in predicting both laminar and turbulent flows. This demonstrates the efficiency of our fluid dynamics simulation method, which adaptively connects new edges based on rewiring delay scores.

Figures 3 and 5 present visualizations of the velocity magnitude contours for two additional datasets. The red mesh indicates high velocity values, while the blue mesh indicates low velocity values. The

Table 1: RMSE results on the Cylinder Flow dataset.

| Method | velocity ($\times 10^{-3}$) | | | pressure ($\times 10^{-3}$) | | |
|---|---|---|---|---|---|---|
| | 1-step | rollout-50 | rollout-all | 1-step | rollout-50 | rollout-all |
| MGN | $2.95 \pm 0.99$ | $9.43 \pm 4.36$ | $53.23 \pm 39.24$ | $97.18 \pm 20.85$ | $26.02 \pm 4.49$ | $11.03 \pm 6.25$ |
| DIGL | $2.64 \pm 1.53$ | $10.50 \pm 6.79$ | $62.35 \pm 40.36$ | $98.62 \pm 22.53$ | $26.47 \pm 5.24$ | $11.47 \pm 5.93$ |
| SDRF | $2.45 \pm 0.54$ | $7.53 \pm 3.52$ | $49.23 \pm 41.93$ | $73.53 \pm 21.76$ | $24.68 \pm 5.63$ | $9.32 \pm 6.16$ |
| BORF | $2.34 \pm 0.12$ | $6.30 \pm 3.70$ | $48.10 \pm 37.20$ | $64.74 \pm 20.82$ | $20.72 \pm 7.52$ | $9.36 \pm 7.95$ |
| PIORF | $1.97 \pm 0.78$ | $7.68 \pm 3.18$ | $47.88 \pm 38.59$ | $57.46 \pm 19.92$ | $19.25 \pm 8.03$ | $7.74 \pm 5.31$ |
| AdaMeshNet | $\mathbf{1.69 \pm 0.56}$ | $\mathbf{5.21 \pm 2.97}$ | $\mathbf{40.37 \pm 38.82}$ | $\mathbf{48.15 \pm 19.48}$ | $\mathbf{12.47 \pm 7.18}$ | $\mathbf{5.86 \pm 4.49}$ |

Table 2: RMSE results on the Airfoil dataset.

| Method | velocity | | | density ($\times 10^{-2}$) | | |
|---|---|---|---|---|---|---|
| | 1-step | rollout-50 | rollout-all | 1-step | rollout-50 | rollout-all |
| MGN | $9.42 \pm 3.13$ | $22.34 \pm 8.39$ | $61.42 \pm 32.35$ | $13.14 \pm 5.13$ | $13.88 \pm 5.93$ | $15.14 \pm 6.49$ |
| DIGL | $9.47 \pm 3.46$ | $20.73 \pm 7.35$ | $63.75 \pm 29.52$ | $11.91 \pm 4.24$ | $12.47 \pm 5.79$ | $14.93 \pm 6.39$ |
| SDRF | $7.09 \pm 2.75$ | $15.24 \pm 3.90$ | $44.25 \pm 41.66$ | $13.30 \pm 4.82$ | $14.93 \pm 5.14$ | $16.38 \pm 5.92$ |
| BORF | $7.51 \pm 3.27$ | $16.33 \pm 2.88$ | $58.24 \pm 28.32$ | $8.01 \pm 1.95$ | $7.91 \pm 3.44$ | $9.81 \pm 4.21$ |
| PIORF | $6.42 \pm 2.25$ | $14.37 \pm 3.95$ | $47.52 \pm 35.48$ | $9.15 \pm 2.20$ | $10.03 \pm 4.39$ | $12.20 \pm 6.13$ |
| AdaMeshNet | $\mathbf{3.25 \pm 1.04}$ | $\mathbf{7.76 \pm 6.25}$ | $\mathbf{28.67 \pm 30.46}$ | $\mathbf{4.98 \pm 2.31}$ | $\mathbf{4.87 \pm 2.47}$ | $\mathbf{7.01 \pm 5.16}$ |

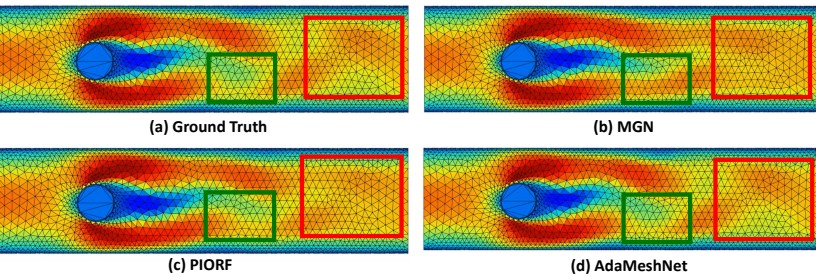

(a) Ground Truth      (b) MGN

(c) PIORF      (d) AdaMeshNet

Figure 3: Velocity magnitude contours on the Cylinder Flow dataset.

red and green boxes in these figures highlight that our method produces velocity contours that are more similar to the ground truth. Specifically, our approach more accurately visualizes the wavelike propagation of velocity to neighboring nodes compared to other methods. This is because our adaptive graph rewiring module more precisely considers inter-particle interactions, allowing it to capture long-range interactions more effectively. Please refer to Section E for more velocity contours.

### 4.3 ABLATION STUDIES

Figure 4 shows the results of ablation studies to examine the effectiveness of our proposed model. Specifically, we perform ablation studies by excluding the distance term $d_{\mathcal{G}}$ in the numerator (i.e., w/o $d_{\mathcal{G}}$), and the velocity difference term $|\mathbf{v}_i - \mathbf{v}_{i^*}|$ in the denominator (i.e., w/o velocity) from Equation 8. We also evaluate the model performance by incorporating the information regarding $d_{\mathcal{G}}$ into the edge weight without including $d_{\mathcal{G}}$ in Equation 8 (i.e., weighted edges). We obtained the following observations: **1)** Excluding $d_{\mathcal{G}}$ and $\mathbf{v}$ from $s_{delay}$ leads to a performance degradation compared to our final model. In particular, removing $d_{\mathcal{G}}$ significantly reduces performance, since the distance information between two nodes is no longer considered when new edges are connected. This indicates that distance information must be sufficiently accounted for when computing the degree of rewiring delay. **2)** Including $d_{\mathcal{G}}$ as an edge weight does not substantially improve performance. This is because, unlike $s_{delay}$, edge weights cannot explicitly consider the rewiring delay. This result highlights that considering temporal delay based on distance information contributes to performance improvement. **3)** The final model with all components included achieves the best performance. This demonstrates that our adaptive rewiring approach, which considers temporal delay and gradual propagation based on both velocity and distance, is the most effective.

### 4.4 HYPERPARAMETER ANALYSIS

In this section, we analyze the sensitivity to the pooling ratio $\alpha$ in Equation 6, which determines the number of edges to be rewired, and the hyperparameter $\beta$ in Equation 8, which represents the influence of distance and velocity.

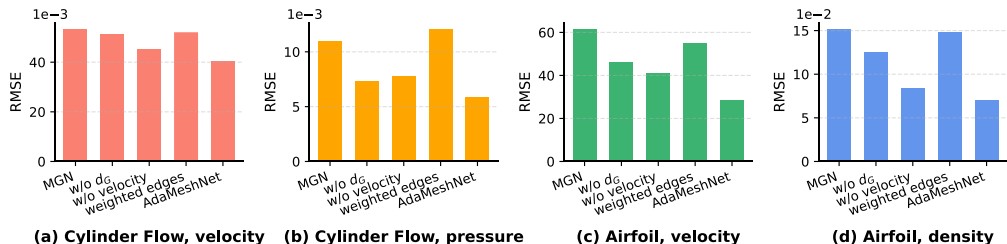

Figure 4: Ablation studies on Cylinder Flow and Airfoil.

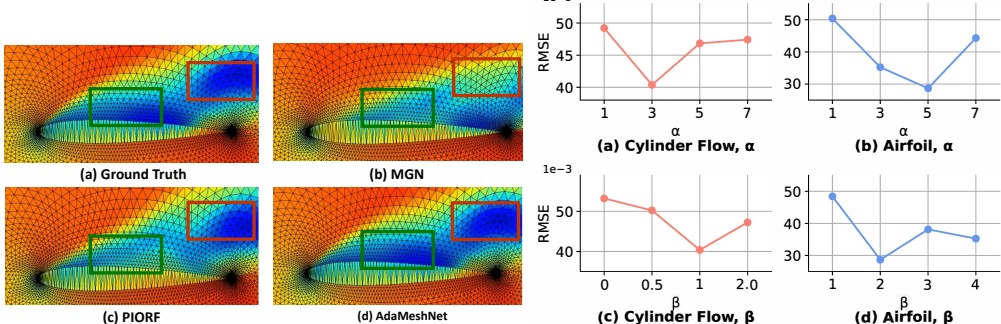

Figure 5: Velocity magnitude contours on Airfoil.

Figure 6: Impact of $\alpha$ and $\beta$.

**Effect of pooling ratio $\alpha$.** Figure 6(a) and (b) show the velocity RMSE for rollout-all over various $\alpha$s. The results show that for the CylinderFlow dataset, the lowest RMSE is achieved at $\alpha = 3\%$, while for the Airfoil dataset, the optimal performance is achieved at $\alpha = 5\%$. These findings indicate that if $\alpha$ is too low, the number of newly rewired nodes is insufficient to effectively capture long-range interactions. Conversely, if $\alpha$ is too high, the model risks losing the original graph topology. This analysis highlights the importance of selecting an optimal $\alpha$ value to balance the preservation of original structure with the ability to capture broader, long-range dependencies. Regarding the training time analysis according to the alpha value, please refer to the Appendix D.

**Effect of hyperparameter $\beta$.** Figure 6(c) and (d) show the velocity RMSE for rollout-all over various $\beta$s. The results indicate that the lowest RMSE is achieved for the Cylinder Flow when $\beta = 1$, while for the Airfoil, the optimal performance is achieved at $\beta = 2$. A lower $\beta$ value places relatively more weight on the influence of velocity than on distance in determining $s_{delay}$, whereas a higher $\beta$ places more weight on distance than on velocity. Airfoil has a wider range of particle velocity values compared to the Cylinder Flow, which can cause the influence of velocity to become overly dominant. To reduce this effect, the optimal $\beta$ is a higher value that increases the influence of distance $d_{\mathcal{G}}$. This demonstrates that the optimal $\beta$ value can be controlled by adjusting the relative influence of velocity and distance, allowing our method to adapt to different graph properties such as velocity distribution.

## 5 CONCLUSION

In this work, we addressed the over-squashing problem inherent in MeshGraphNets (MGN) for fluid dynamics simulations by introducing AdaMeshNet, a novel adaptive graph rewiring framework. Unlike previous static rewiring methods that treat distant nodes as immediate neighbors, our approach adaptively rewires edges during the message-passing process, considering the gradual propagation of physical interactions. We propose a new rewiring delay score based on velocity difference and inter-node distance. This score determines the layer at which new edges are added, allowing our model to more realistically simulate the time-delayed effects of long-range interactions. Experimental results confirm that AdaMeshNet outperforms existing static rewiring methods, and our visualizations highlight its superior ability to accurately capture complex flow phenomena. This work represents a significant step forward in developing more accurate and physically-grounded GNNs for computational fluid dynamics.

## ETHIC STATEMENT

This research complies with the ICLR Code of Ethics. All experiments were performed on publicly accessible and widely adopted benchmark datasets, which contain no personally identifiable or sensitive information, thereby minimizing privacy concerns. Our research is intended to enhance application in real-world fluid dynamics without enabling harmful uses or misuse. We are dedicated to maintaining scientific integrity and provide anonymized source code to guarantee transparency and reproducibility. After careful consideration of the potential impacts of this work, we conclude that it does not present notable ethical concerns.

## REPRODUCIBILITY STATEMENT

We provide all the necessary details in Section 4 and the Appendix to ensure the reproducibility of our study. In addition, our source code is available at `https://anonymous.4open.science/r/AdaMeshNet-9321`.

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

# A  DETAILED PROOF OF LEMMA 1

The following equations describe the message-passing scheme used in MGN:

$$\mathbf{e}_{ij}^{(r)} = f_E\left(\mathbf{e}_{ij}^{(r-1)}, \mathbf{h}_i^{(r-1)}, \mathbf{h}_j^{(r-1)}\right),\tag{14}$$

$$\mathbf{h}_i^{(r)} = f_V\left(\mathbf{h}_i^{(r-1)}, \sum_{j=1}^n \hat{A}_{ij}\mathbf{e}_{ij}^{(r)}\right) = f_V\left(\mathbf{h}_i^{(r-1)}, \sum_{j=1}^n \hat{A}_{ij}f_E\left(\mathbf{e}_{ij}^{(r-1)}, \mathbf{h}_i^{(r-1)}, \mathbf{h}_j^{(r-1)}\right)\right).\tag{15}$$

Based on the above equations, we can expand their derivatives as follows:

$$\frac{\partial \mathbf{h}_i^{(r)}}{\partial \mathbf{x}_s} = \frac{\partial f_V}{\partial \mathbf{h}_i^{(r-1)}}\frac{\partial \mathbf{h}_i^{(r-1)}}{\partial \mathbf{x}_s} + \frac{\partial f_V}{\partial \mathbf{z}_i^{(r)}}\sum_j \hat{A}_{ij}\frac{\partial \mathbf{e}_{ij}^{(r)}}{\partial \mathbf{x}_s},\tag{16}$$

$$\frac{\partial \mathbf{e}_{ij}^{(r)}}{\partial \mathbf{x}_s} = \frac{\partial f_E}{\partial \mathbf{e}_{ij}^{(r-1)}}\frac{\partial \mathbf{e}_{ij}^{(r-1)}}{\partial \mathbf{x}_s} + \frac{\partial f_E}{\partial \mathbf{h}_i^{(r-1)}}\frac{\partial \mathbf{h}_i^{(r-1)}}{\partial \mathbf{x}_s} + \frac{\partial f_E}{\partial \mathbf{h}_j^{(r-1)}}\frac{\partial \mathbf{h}_j^{(r-1)}}{\partial \mathbf{x}_s}.\tag{17}$$

where $\mathbf{z}_i^{(r)} = \sum_{j=1}^n \hat{A}_{ij}\mathbf{e}_{ij}^{(r)}$.

First, to derive an upper bound of $\left|\frac{\partial \mathbf{h}_i^{(r)}}{\partial \mathbf{x}_s}\right|$, we plug Equation 17 into Equation 16 and obtain the following expression:

$$\frac{\partial \mathbf{h}_i^{(r)}}{\partial \mathbf{x}_s} = \frac{\partial f_V}{\partial \mathbf{h}_i^{(r-1)}}\frac{\partial \mathbf{h}_i^{(r-1)}}{\partial \mathbf{x}_s} + \frac{\partial f_V}{\partial \mathbf{z}_i^{(r)}}\sum_j \hat{A}_{ij}\left(\frac{\partial f_E}{\partial \mathbf{e}_{ij}^{(r-1)}}\frac{\partial \mathbf{e}_{ij}^{(r-1)}}{\partial \mathbf{x}_s} + \frac{\partial f_E}{\partial \mathbf{h}_i^{(r-1)}}\frac{\partial \mathbf{h}_i^{(r-1)}}{\partial \mathbf{x}_s} + \frac{\partial f_E}{\partial \mathbf{h}_j^{(r-1)}}\frac{\partial \mathbf{h}_j^{(r-1)}}{\partial \mathbf{x}_s}\right).\tag{18}$$

Note that $s$ is an $r$-hop neighbor of $i$, while $\mathbf{h}_i^{(r-1)}$, $\mathbf{h}_j^{(r-2)}$, and $\mathbf{e}_{ij}^{(r-1)}$ are embeddings made by aggregating the information from up to $(r-1)$-hop neighbors of $i$. Thus, the Jacobians $\frac{\partial \mathbf{h}_i^{(r-1)}}{\partial \mathbf{x}_s}$, $\frac{\partial \mathbf{h}_j^{(r-2)}}{\partial \mathbf{x}_s}$ and $\frac{\partial \mathbf{e}_{ij}^{(r-1)}}{\partial \mathbf{x}_s}$ are zero matrices, and this enables us to recursively expand $\frac{\partial \mathbf{h}_i^{(r)}}{\partial \mathbf{x}_s}$ as follows:

$$\begin{aligned}\frac{\partial \mathbf{h}_i^{(r)}}{\partial \mathbf{x}_s} &= \sum_j \hat{A}_{ij}\frac{\partial f_V}{\partial \mathbf{z}_i^{(r)}}\frac{\partial f_E}{\partial \mathbf{h}_j^{(r-1)}}\frac{\partial \mathbf{h}_j^{(r-1)}}{\partial \mathbf{x}_s}\\ &= \sum_j \hat{A}_{ij}\frac{\partial f_V}{\partial \mathbf{z}_i^{(r)}}\frac{\partial f_E}{\partial \mathbf{h}_j^{(r-1)}}\sum_k \hat{A}_{jk}\frac{\partial f_V}{\partial \mathbf{z}_j^{(r-1)}}\frac{\partial f_E}{\partial \mathbf{h}_k^{(r-2)}}\frac{\partial \mathbf{h}_k^{(r-2)}}{\partial \mathbf{x}_s}\\ &= \cdots = \sum_{j_1,\ldots,j_r} \hat{A}_{ij_1}\hat{A}_{j_1 j_2}\cdots \hat{A}_{j_{r-1}j_r}\cdot J_{ij_1\cdots j_r}(X)\cdot \frac{\partial \mathbf{h}_{j_r}^{(0)}}{\partial \mathbf{x}_s},\end{aligned}\tag{19}$$

where $J_{ij_1\cdots j_r}(X)$ represents the product of $r$ second partial derivatives of $f_V$ and $r$ third partial derivatives of $f_E$ with $j_l$ indicating the index of $i$'s $l$-hop neighbors. Since $\partial_{\mathbf{x}_s}\mathbf{h}_{j_r}^{(0)} = \partial_{\mathbf{x}_s}\mathbf{x}_{j_r} = \delta_{j_r s}$ holds, we obtain

$$\frac{\partial \mathbf{h}_i^{(r)}}{\partial \mathbf{x}_s} = \sum_{j_1,\ldots,j_{r-1}} \hat{A}_{ij_1}\hat{A}_{j_1 j_2}\cdots \hat{A}_{j_{r-1}s}\cdot J_{ij_1\cdots j_{r-1}s}(X)\tag{20}$$

Finally, since $\left|J_{ij_1\cdots j_{r-1}s}(X)\right| \le (\alpha_e\beta_h)^r$ holds by the given assumptions, we obtain

$$\begin{aligned}\left|\frac{\partial \mathbf{h}_i^{(r)}}{\partial \mathbf{x}_s}\right| &\le \sum_{j_1,\ldots,j_{r-1}} \hat{A}_{ij_1}\hat{A}_{j_1 j_2}\cdots \hat{A}_{j_{r-1}s}(\alpha_e\beta_h)^r\\ &= (\alpha_e\beta_h)^r\left(\hat{A}^r\right)_{is}.\end{aligned}\tag{21}$$

Second, to derive an upper bound of $\left| \frac{\partial \mathbf{e}_{ij}^{(r)}}{\partial \mathbf{x}_s} \right|$. To this end, we plug Equation 16 into Equation 17 and obtain the following expression:

$$\frac{\partial \mathbf{e}_{ij}^{(r)}}{\partial \mathbf{x}_s} = \frac{\partial f_E}{\partial \mathbf{e}_{ij}^{(r-1)}} \frac{\partial \mathbf{e}_{ij}^{(r-1)}}{\partial \mathbf{x}_s} + \frac{\partial f_E}{\partial \mathbf{h}_i^{(r-1)}} \frac{\partial \mathbf{h}_i^{(r-1)}}{\partial \mathbf{x}_s} + \frac{\partial f_E}{\partial \mathbf{h}_j^{(r-1)}} \left( \frac{\partial f_V}{\partial \mathbf{h}_j^{(r-2)}} \frac{\partial \mathbf{h}_j^{(r-2)}}{\partial \mathbf{x}_s} + \frac{\partial f_V}{\partial \mathbf{z}_j^{(r-1)}} \sum_k \hat{A}_{jk} \frac{\partial \mathbf{e}_{jk}^{(r-1)}}{\partial \mathbf{x}_s} \right).$$
(22)

As mentioned above, the Jacobians $\frac{\partial \mathbf{h}_i^{(r-1)}}{\partial \mathbf{x}_s}$, $\frac{\partial \mathbf{h}_j^{(r-2)}}{\partial \mathbf{x}_s}$ and $\frac{\partial \mathbf{e}_{ij}^{(r-1)}}{\partial \mathbf{x}_s}$ are zero matrices, and this enables us to recursively expand $\frac{\partial \mathbf{e}_{ij}^{(r)}}{\partial \mathbf{x}_s}$ as follows:

$$\begin{aligned}
\frac{\partial \mathbf{e}_{ij}^{(r)}}{\partial \mathbf{x}_s} &= \frac{\partial f_E}{\partial \mathbf{h}_j^{(r-1)}} \frac{\partial f_V}{\partial \mathbf{z}_j^{(r-1)}} \sum_k \hat{A}_{jk} \frac{\partial \mathbf{e}_{jk}^{(r-1)}}{\partial \mathbf{x}_s} \\
&= \frac{\partial f_E}{\partial \mathbf{h}_j^{(r-1)}} \frac{\partial f_V}{\partial \mathbf{z}_j^{(r-1)}} \sum_k \hat{A}_{jk} \frac{\partial f_E}{\partial \mathbf{h}_k^{(r-2)}} \frac{\partial f_V}{\partial \mathbf{z}_k^{(r-2)}} \sum_m \hat{A}_{km} \frac{\partial \mathbf{e}_{km}^{(r-1)}}{\partial \mathbf{x}_s} \\
&= \cdots = \sum_{j_2,\dots,j_r} \hat{A}_{jj_2} \hat{A}_{j_2 j_3} \cdots \hat{A}_{j_{r-1} j_r} \cdot J_{jj_2\cdots j_{r-1}}(X) \cdot \frac{\partial \mathbf{e}_{j_{r-1} j_r}^{(1)}}{\partial \mathbf{x}_s} \\
&= \sum_{j_2,\dots,j_r} \hat{A}_{jj_2} \hat{A}_{j_2 j_3} \cdots \hat{A}_{j_{r-1} j_r} \cdot J_{jj_2\cdots j_{r-1}}(X) \cdot \frac{\partial f_E}{\partial \mathbf{h}_{j_r}^{(0)}} \frac{\partial \mathbf{h}_{j_r}^{(0)}}{\partial \mathbf{x}_s} \\
&= \sum_{j_2,\dots,j_r} \hat{A}_{ij_2} \hat{A}_{j_2 j_3} \cdots \hat{A}_{j_{r-1} j_r} \cdot J_{jj_2\cdots j_r}(X) \cdot \frac{\partial \mathbf{h}_{j_r}^{(0)}}{\partial \mathbf{x}_s},
\end{aligned}$$
(23)

where $J_{jj_2\cdots j_r}(X)$ represents the product of $r-1$ second partial derivatives of $f_V$ and $r$ third partial derivatives of $f_E$ with $j_l$ indicating the index of $i$'s $l$-hop neighbors. Since $\partial_{\mathbf{x}_s} \mathbf{h}_{j_r}^{(0)} = \partial_{\mathbf{x}_s} \mathbf{x}_{j_r} = \delta_{j_r s}$ holds, we obtain

$$\frac{\partial \mathbf{e}_{ij}^{(r)}}{\partial \mathbf{x}_s} = \sum_{j_2,\dots,j_{r-1}} \hat{A}_{jj_2} \hat{A}_{j_2 j_3} \cdots \hat{A}_{j_{r-1} s} \cdot J_{jj_2\cdots j_{r-1} s}(X)$$
(24)

Finally, since $\left| J_{ij_2\cdots j_{r-1} s}(X) \right| \leq \alpha_e^{r-1} \beta_h^r$ holds by the given assumptions, we obtain

$$\begin{aligned}
\left| \frac{\partial \mathbf{e}_{ij}^{(r)}}{\partial \mathbf{x}_s} \right| &\leq \sum_{j_2,\dots,j_{r-1}} \hat{A}_{jj_2} \hat{A}_{j_2 j_3} \cdots \hat{A}_{j_{r-1} s} \alpha_e^{r-1} \beta_h^r \\
&= \alpha_e^{r-1} \beta_h^r \left( \hat{A}^{r-1} \right)_{js}.
\end{aligned}$$
(25)

## B   RELATED WORK

### B.1   HIGH-DIMENSIONAL PHYSICS MODELS

Deep learning-based modeling for high-dimensional physics problems has been actively used in fluid dynamics (Bhatnagar et al., 2019; Zhang et al., 2018; Guo et al., 2016). Compared to complex Finite Element Methods (FEM), deep learning-based approaches offer efficient execution times (Um et al., 2018; Xie et al., 2018; Wiewel et al., 2019) and can be applied in real-world physical environments where all parameters are not fully known (De Bézenac et al., 2018). Domain-specific loss functions (Lee & You, 2019; Wang et al., 2020) or feature normalization that incorporates physical knowledge (Thuerey et al., 2020) can help improve the performance of deep learning models.

All the methods mentioned above use regular grid-based convolutions to model high-dimensional physics problems. Holden et al. (2019) applied Principal Component Analysis (PCA) to cloth data to reduce the dimensionality of high-dimensional systems and then performed simulations in the reduced-dimensional space. Recent studies (Li et al., 2019; Sanchez-Gonzalez et al., 2020) have utilized Graph Neural Networks (GNNs) to model physics systems such as fluid simulations. Conventional FEM requires complex calculations and struggles to find accurate solutions when dealing with nonlinear problems. In contrast, GNN-based methods can predict nonlinear problems more quickly and accurately by learning these complex, nonlinear relationships directly from data (Luo et al., 2018).

### B.2   GRAPH REWIRING METHODS

Mesh refinement techniques (Löhner, 1995; Liu et al., 2022), which adaptively create high-resolution meshes, can exacerbate the over-squashing problem. This leads to a loss of information as long-range information is compressed into a fixed-size feature vector. To solve this problem, various methods have been attempted to address over-squashing in GNNs (Fesser & Weber, 2023; Shi et al., 2023; Finkelshtein et al., 2023; Barbero et al., 2024; Errica et al., 2023; Tortorella & Micheli, 2022). To address this, various graph rewiring techniques have been proposed. Gasteiger et al. (2019) introduced new edges based on diffusion distance to induce a smoother adjacency matrix. However, this method is not suitable for tasks that require connecting long diffusion distances. Topping et al. (2022) detects nodes with negative curvature and adds new edges from these nodes. (Karhadkar et al., 2023) enhances the efficiency of information transfer by connecting edges that maximize the spectral gap. Nguyen et al. (2023) propose connecting new edges based on the Ollivier-Ricci curvature, which is designed to mitigate both over-smoothing and over-squashing simultaneously. Attali et al. (2024) connect nodes based on Delaunay triangulation to make connections regular and uniform, preventing information from being excessively concentrated on specific nodes. However, since mesh-based simulations are already constructed with a regular grid-like structure similar to triangulation, Delaunay triangulation offers little additional benefit to mesh graphs. All of these studies employ a static approach, completing all rewiring before applying the GNN. Our method adaptively rewires new edges during the message-passing process, considering the progressive propagation of physical interactions.

## C   DATASETS

In this paper, we used the Cylinder Flow and Airfoil datasets, which are commonly used in fluid simulations. Cylinder Flow represents a laminar flow model, where the fluid moves smoothly and regularly, whereas Airfoil represents a turbulent flow model, where the fluid moves irregularly and chaotically.

### C.1   CYLINDER FLOW

The Cylinder Flow dataset contains physical quantities of a fluid as it flows around a cylinder. This model has practical applications in various industrial fields, particularly in environments involving cylindrical pipes. The model can predict how fluid flow patterns change depending on the size and position of the cylinder. This prediction ability can contribute to solving real-world engineering problems, such as designing cooling systems or improving fluid transportation efficiency. The dataset includes 1,000 flow results, each with 600 time steps.

## C.2 AIRFOIL

The Airfoil dataset includes physical quantities related to fluid flow around an aircraft wing. It contains complex turbulent phenomena, which helps our model learn to handle diverse flow conditions. An aircraft wing has a special cross-sectional shape called an airfoil. This shape causes air to flow over and under the wing at different speeds, and this velocity difference generates lift, which is the key force that allows an airplane to fly. This dataset is crucial for designing and validating the performance of wings in various aerospace applications, such as airplanes and helicopters. Specifically, the model can be used to predict how airflow changes around a wing and how this affects the stability of the aircraft. The Airfoil dataset also includes 1,000 flow simulations, each with 600 time steps.

## D  TRAINING TIME ANALYSIS

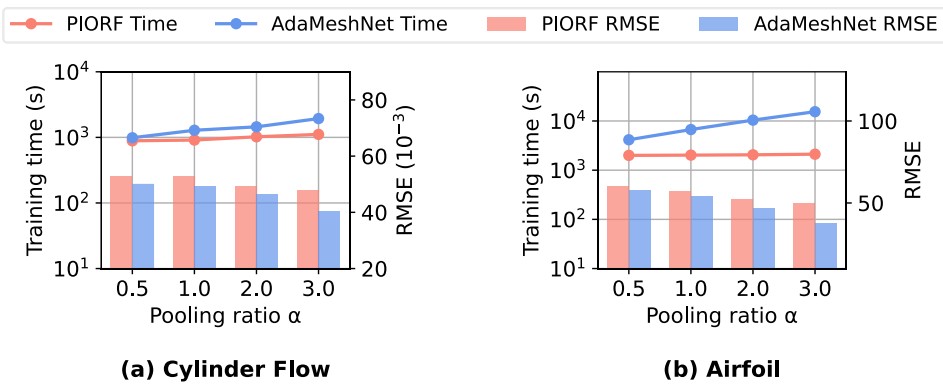

(a) Cylinder Flow                    (b) Airfoil

Figure 7: Time efficiency on Cylinder Flow and Airfoil.

In this section, we measure the training time for mesh simulation to analyze the time efficiency. We compare the training time of our model with PIORF, the most time-efficient static rewiring method. Figure 7 shows the training time over various pooling ratios $\alpha$ on Cylinder Flow and Airfoil datasets. According to Figure 7, our AdaMeshNet model takes longer training time compared to the existing PIORF model, since it involves calculating the rewiring delay score during the message-passing process. Nevertheless, the result shows that as the pooling ratio $\alpha$ decreases, the training time of AdaMeshNet becomes comparable to that of PIORF. While AdaMeshNet is somewhat less efficient in terms of training time compared to PIORF, the bar graphs in Figure 7 show that it provides a significant advantage in terms of improved prediction accuracy. In real-world fluid dynamics simulations, even a small difference in accuracy can have a substantial impact on the overall reliability of the model, which makes a slight increase in training time acceptable. For instance, the Airfoil dataset can be used to design and validate wing performance. In the aerospace field, the performance of the wing is closely related to safety, making improvements in accuracy much more important than training time efficiency. Therefore, even with a slight increase in training time, our model, which significantly contributes to improving accuracy, is expected to have high applicability to real-world problems in fluid dynamics. In conclusion, while AdaMeshNet takes longer to train compared to PIORF, the extra time is spent on modeling the gradual propagation we propose, which can be seen as a reasonable cost to mimic more realistic models in complex fluid simulations.

# E OTHER VELCITY CONTOURS

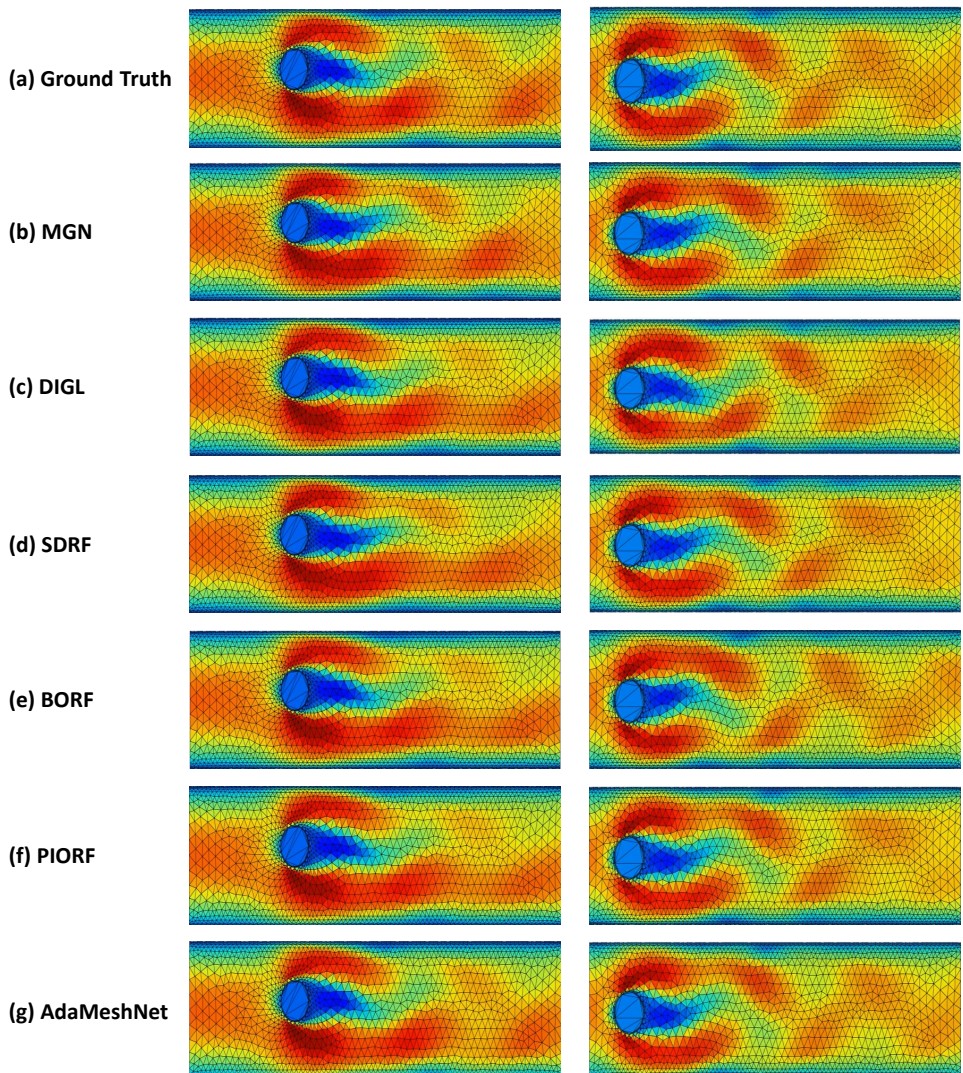

Figure 8: Other Velcity Contours

# F  ALGORITHM

---

**Algorithm 1:** Adaptive Graph Rewiring for Mesh-Based GNN Training

---

**Input** : Training mesh $\mathcal{M}_t$
**Output:** Updated mesh $\mathcal{M}_{t+1}$

1 **for** *epoch = 1, 2, ..., T* **do**
2     **Preprocessing: for** *node $v_i$ in $\mathcal{M}_t$* **do**
3        Calculate node curvature $\gamma_i$ using Eq. 5
4     **end**
5     Identify bottleneck nodes $\mathcal{V}_{\text{lowORC}}$ using Eq. 6
6     **for** *each $v_i \in \mathcal{V}_{lowORC}$* **do**
7        Select optimal connection node $v_{i*}$ using Eq. 7
8        Calculate rewiring delay score $s_{\text{delay}}(i, i^*)$ using Eq. 8
9     **end**
10     **Encoder: for** *each node $v_i$ and edge $e_{ij}$ in $\mathcal{M}_t$* **do**
11        Calculate node embedding $\mathbf{h}_i$ using Eq. 9
12        Calculate edge embedding $\mathbf{e}_{ij}$ using Eq. 9
13     **end**
14     **Processor: for** *layer l = 0, 1, ..., L-1* **do**
15        **for** *each $v_i \in \mathcal{V}_{lowORC}$* **do**
16           **for** *each optimal connection node $v_{i*}$* **do**
17              If $l < s_{\text{delay}}(i, i^*) \leq l + 1$ Add $v_{i*}$ to neighbor set $\mathcal{N}_i^{l+1}$
18           **end**
19        **end**
20        **for** *each node $v_i$ in $\mathcal{M}_t$* **do**
21           Initialize neighbor set $\mathcal{N}_i^0$ as direct neighbors from $\mathcal{E}$
22           **for** *each node $v_j \in \mathcal{N}_i^{l+1}$* **do**
23              Update edge embedding $\mathbf{e}_{ij}^{l+1}$ using Eq. 11
24              Update node embedding $\mathbf{h}_i^{l+1}$ using Eq. 12
25           **end**
26        **end**
27     **end**
28     **Decoder and State Updater: for** *each node $v_i$ in $\mathcal{V}$* **do**
29        Compute the predicted state $\hat{q}_i^{t+1}$ using Eq. 13
30     **end**
31     **Update Mesh:** Update mesh $\mathcal{M}_{t+1}$ based on the updated nodes $\mathcal{V}$ and their corresponding states
32 **end**

---

# G  NOTATIONS

In this section, we summarize the main notations used in this paper. Table 3 provides the main notation and their descriptions.

Table 3: Summary of the main notations used in this paper.

| Notation | Description |
|---|---|
| $\mathcal{G} = (\mathcal{V}, \mathcal{E})$ | A graph with a set of nodes $\mathcal{V}$ and a set of edges $\mathcal{E}$ |
| $n = |\mathcal{V}|$ | Total number of nodes |
| $\mathcal{N}_i$ | Set of neighbors for node $i$ |
| $\mathbf{x}_i \in \mathbb{R}^{p_0}$ | Initial feature vector of node $i$ |
| $\mathbf{v}_i$ | Velocity vector of node $i$ |
| $d_\mathcal{G}(i, j)$ | Shortest path distance between nodes $i$ and $j$ in graph $\mathcal{G}$ |
| $l$ | Layer index of the GNN |
| $L$ | Total number of message-passing blocks (layers) |
| $\mathbf{h}_i^{(l)}$ | Hidden representation (embedding) of node $i$ at layer $l$ |
| $\mathbf{e}_{ij}^{(l)}$ | Hidden representation (embedding) of edge $(i, j)$ at layer $l$ |
| $f_V$ | Node update function (MLP) |
| $f_E$ | Edge update function (MLP) |
| $r$ | Distance between two nodes in hops |
| $B_r(i)$ | Set of nodes within $r$ hops from node $i$ (receptive field) |
| $\partial \mathbf{h}_i^{(r)} / \partial \mathbf{x}_s$ | Jacobian of the hidden representation of node $i$ at layer $r$ w.r.t. the input feature of node $s$ |
| $\hat{A}$ | Normalized augmented adjacency matri |
| $\alpha_e$ | Upper bounds for the second partial derivatives of $f_V$ |
| $\beta_h$ | Upper bounds for the third partial derivatives $f_E$ |
| $\kappa(i, j)$ | Ollivier-Ricci Curvature of the edge $(i, j)$ |
| $\gamma_i$ | Average curvature of node $i$ (local geometric information) |
| $\mathcal{V}_{\text{lowORC}}$ | Set of bottleneck nodes in the bottom $a\%$ of curvature |
| $v_{i^*}$ | Optimal node to be rewired with the bottleneck node $v_i$ |
| $s_{\text{delay}}(i, i^*)$ | Rewiring delay score for the edge $(i, i^*)$ |
| $\beta$ | Hyperparameter used in calculating the delay score |
| $\mathcal{N}_i^l$ | Set of neighbors for node $i$ at layer $l$ (with rewiring applied) |
| $\hat{\mathbf{v}}_i$ | Predicted velocity gradient of node $i$ from the decoder |
| $\hat{\mathbf{v}}_i^{t+1}$ | Predicted velocity of node $i$ at time $t + 1$ |

