# OpenReview forum: "Adaptive Graph Rewiring to Mitigate Over-Squashing in Mesh-Based GNNs for Fluid Dynamics Simulations"
_ICLR.cc/2026/Conference — ICLR 2026 Conference Withdrawn Submission_

### Official Review · Reviewer_gHyn · 2025-10-31

**Soundness:** 3
**Presentation:** 3
**Contribution:** 2
**Rating:** 2
**Confidence:** 4

**Summary:**

The authors propose AdaMeshNet, a framework that introduces adaptive graph rewiring during the message-passing process, as opposed to static rewiring methods that perform all rewiring before GNN training. The method identifies bottleneck nodes using Ollivier-Ricci curvature and computes a rewiring delay score based on shortest-path distance and velocity difference. Experiments on CylinderFlow and Airfoil datasets demonstrate improved prediction accuracy for physical quantities like velocity and pressure compared to static rewiring baselines.

**Strengths:**

1. The proposed AdaMeshNet introduces dynamic, layer-dependent rewiring via a rewiring delay score based on velocity difference and graph distance. This adaptive process is conceptually distinct from static rewiring methods, and better reflects gradual physical propagation.
2. The visualization and analysis (e.g., Cylinder Flow velocity contour comparisons) provide intuitive physical explanations of how adaptive rewiring reduces unrealistic instantaneous information transfer.

**Weaknesses:**

1. The evaluation is restricted to two fluid datasets (CylinderFlow and Airfoil). Broader validation on non-fluid domains (e.g., deformable solids, elastic plates, cloth dynamics) would better demonstrate generality.
2. The key rewiring criterion — connecting nodes with large velocity differences — is directly inherited from PIORF (Yu et al., 2025), potentially diminishing the novelty of the contribution. I suggest investigating other physical quantities (e.g., pressure, density, or vorticity) that might be more appropriate for different physical systems. For example, pressure and density gradients dominate the dynamics in compressible fluids, whereas in deformable solids or elastic media, strain or stress fields may play a more important role. Limiting the rewiring signal to velocity may be suboptimal or even physically inconsistent in other regimes.
3. While the delay-based rewiring is motivated by gradual propagation in fluids, it remains unclear how the same mechanism adapts to physics where interaction speed or propagation modality differs (e.g., elastic or collision dynamics).
4. Prior works [1–5] employ hierarchical or multiscale connectivity learning to address long-range information loss. Discussion or comparison with these approaches would help clarify how AdaMeshNet differs from such hierarchical designs.
5. The paper does not analyze the computational cost of dynamic rewiring and curvature computation. Reporting runtime and memory overhead relative to static baselines would improve completeness.

[1] EvoMesh: Adaptive Physical Simulation with Hierarchical Graph Evolutions. ICML 2025

[2] Eagle: Large-Scale Learning of Turbulent Fluid Dynamics with Mesh Transformers. ICLR 2023

[3] Efficient Learning of Mesh-Based Physical Simulation with BSMS-GNN. ICML 2023

[4] Physics meets Topology: Physics-informed topological neural networks for learning rigid body dynamics

[5] Learning Flexible Body Collision Dynamics with Hierarchical Contact Mesh Transformer. ICLR 2024

**Questions:**

Please refer to Weakness section.

---

### Official Review · Reviewer_ova9 · 2025-10-31

**Soundness:** 3
**Presentation:** 3
**Contribution:** 3
**Rating:** 4
**Confidence:** 3

**Summary:**

This paper addresses the over-squashing problem in mesh-based Graph Neural Networks used for fluid dynamics simulations. The authors propose AdaMeshNet, a framework that performs adaptive graph rewiring during message passing rather than beforehand. The method uses Ollivier–Ricci curvature to identify bottleneck nodes where information flow is restricted, and computes a rewiring delay score based on the graph distance and velocity difference between nodes. This score determines at which message-passing layer a new edge is added, allowing long-range interactions to be incorporated gradually, in a manner consistent with physical propagation. Experiments on two fluid dynamics datasets show that AdaMeshNet outperforms baselines in predicting key physical quantities such as velocity, pressure, and density, achieving the lowest RMSEs.

**Strengths:**

- The paper introduces a layer-wise, physically grounded rewiring approach, addressing the unrealistic assumption of instantaneous interactions in static methods.
- Experiment results demonstrate that the proposed method has superior performance.

**Weaknesses:**

- Experiments are confined to two datasets; broader validation would strengthen generality claims.
- Adaptive rewiring during training increases computational overhead. It would be better if the authors could add some runtime or scalability analysis.
- Performance depends notably on $\alpha$ (rewiring ratio) and $\beta$ (distance–velocity weighting), but the tuning process may not generalize easily to new settings.
- Lemma 1 formalizes information decay but does not quantify how adaptive rewiring modifies this bound.

**Questions:**

The paper claims that AdaMeshNet models the gradual propagation of physical interactions by introducing rewiring delays across message-passing layers. However, in physical fluid systems, the delay of interaction propagation is a temporal phenomenon occurring over simulation time steps. Could the authors clarify the reason for modeling this temporal propagation in message-passing layers？

---

### Official Review · Reviewer_RkBo · 2025-11-01

**Soundness:** 2
**Presentation:** 2
**Contribution:** 1
**Rating:** 2
**Confidence:** 4

**Summary:**

Here is a concise review based on my analysis.
The paper proposes AdaMeshNet, a GNN framework for fluid dynamics that aims to mitigate over-squashing by introducing an "adaptive rewiring" mechanism. This mechanism delays the addition of new edges to specific message-passing layers based on a "rewiring delay score."

While the paper addresses a relevant problem and shows slight empirical improvements, I must recommend **Reject**.

The work suffers from three critical flaws:
1.  The core physics-informed rewiring strategy (identifying bottlenecks with Ollivier-Ricci Curvature and connecting them to nodes with maximal velocity difference) is not novel and is adopted directly from prior art, PIORF. The only novelty is an algorithmic tweak to *delay* this connection.
2.  The entire justification for this delay is that instantaneous interactions are "physically wrong". This is fundamentally incorrect for the paper's primary benchmark, CylinderFlow which models *incompressible* flow. In this regime, information (pressure) propagates *instantaneously* by definition.
3.  The core mechanism, the "rewiring delay score" (Eq. 8), is dimensionally incoherent. It compares a unitless GNN layer index $l$ to a score $s_{delay}$ that, by its formulation (unitless distance divided by velocity), must have physical units ($Time/Length$). This comparison is physically and mathematically meaningless.

The paper is an incremental work, while it's intuition is physically unjustified, and mathematically flawed.

**Strengths:**

1.  The paper tackles the important and challenging problem of over-squashing in mesh-based GNNs for physics simulations.
2.  The empirical results in Tables 1 and 2 consistently show that AdaMeshNet achieves a lower RMSE than the baselines, including PIORF, across both datasets.
3.  The ablation study effectively demonstrates that both the distance and velocity components of the heuristic score (Eq. 8) are necessary to achieve the reported performance.

**Weaknesses:**

1.  The paper presents its physics-informed selection strategy as novel. However, the entire logic—(1) identify topological bottlenecks using node-level ORC, and select a rewiring target $v_{i^*}$ that maximizes the velocity difference $||v_i - v_j||$—is identical to the strategy proposed in the PIORF paper[1]; The *only* novel contribution of this work is the calculation of $s_{delay}$ to decide *when* to add this pre-selected edge, which is an incremental modification, not a "novel framework".

2. The paper's central premise is that modeling the "gradual propagation of physical interactions" is more realistic than assuming "instantaneous interactions". This justification is in direct conflict with the physics of the paper's main benchmark. The CylinderFlow dataset models incompressible flow. In an incompressible fluid, the speed of sound is infinite, and thus information (propagated via the pressure field) is, by definition, "sensed instantaneously at all other points in the fluid". The "flaw" this paper claims to fix is, in fact, the *correct* physical assumption for this system. The authors appear to be conflating the *advection* of mass (which takes time) with the *propagation* of information (which is instantaneous in this model).

3.  The paper's core mechanism, Eq. 8, does not make sense dimensionally speaking, it's using some dimensionless quantity to divide realistic velocity. Also, the usage of geodesic minimal (edge-count) distance does not make sense too, as for FVM (finite volume method) simulation, the velocity is never defined on the edge.

4.  The paper provides a standard definition for ORC, which involves computing the $W_1$ distance. This is known to be computationally expensive, requiring the solution of a linear programming problem for each edge. The only exception is for bi-partite graph or graph with girth >=5 [2]. But here we have triangular mesh (girth=3). PIORF performs this expensive calculation *once* as a preprocessing step. In contrast, AdaMeshNet's Algorithm 1 places the *entire* preprocessing step—including the ORC calculation and the $O(N^2)$ search for optimal partners—*inside* the main training loop, to be run *every epoch*. This suggests a prohibitive increase in computational cost, however the paper does not report implementation details or timing.

[1] https://openreview.net/forum?id=qkBBHixPow
[2] https://sites.stat.columbia.edu/sumitm/Ricci.pdf

**Questions:**

1.  How do the authors justify the core premise of "gradual propagation" when the CylinderFlow dataset models incompressible flow, where pressure propagation is instantaneous?
2.  Can the authors provide a dimensional analysis for Equation 8? How can the unitless layer index $l$ be mathematically compared to $s_{delay}$, which (based on its components) has units of $Time/Length$?
3.  The calculation of $W_1$ for ORC on triangular meshes is non-trivial and requires solving a linear program. How is this step practically implemented? Is a numerical optimizer used, and if so, why is this extremely expensive step placed *inside* the main training loop (Algorithm 1) rather than as a one-time preprocessing step? Can you list the method you used? Can you also list the timing?

---

### Official Review · Reviewer_9syx · 2025-11-01

**Soundness:** 2
**Presentation:** 2
**Contribution:** 3
**Rating:** 4
**Confidence:** 4

**Summary:**

This paper introduces AdaMeshNet, an adaptive graph rewiring framework for mesh-based graph neural networks (GNNs) applied to fluid dynamics simulations. The work is devoted to solution of the over-squashing problem in message-passing GNNs, which becomes especially severe when mesh refinement increases node density in regions with strong gradients. Existing static rewiring approaches (DIGL, SDRF, BORF, PIORF) add all new edges before training, leading to unphysical “instantaneous” long-range interactions.

AdaMeshNet proposes a dynamic layer-wise rewiring process that models the gradual propagation of physical interactions:
* bottleneck nodes are identified using Ollivier–Ricci curvature
* for each bottleneck, an optimal connection node is selected based on velocity difference and time delay with which the information will reach the bottleneck

The model is implemented on top of MeshGraphNet (MGN) and evaluated on Cylinder Flow and Airfoil datasets. Results show that AdaMeshNet outperforms all static rewiring baselines (DIGL, SDRF, BORF, PIORF). However the big stds of the obtained metrics indicate the unstable model performance.

**Strengths:**

* physically consistent rewiring: introduces an adaptive rewiring mechanism that reflects real-world gradual propagation of information in fluids and considers both curvature and velocity/distance
* theoretical grounding: provides a mathematical analysis of over-squashing in MGN, supported by some Jacobian-based derivation
* dynamic adaptation
* ablation studies are present

**Weaknesses:**

* the computational efficiency is not clear
* the provided code is just a placeholder (message "File not found" appears when trying to see it)
* looking to Tables 1 and 2 I observe the stds of metrics comparable to the metrics values, what means unstable training
* ablation studies could be better if errors of the obtained metrics would have also plotted

**Questions:**

* Can the model be used to model the multiphase flow?
* Does your approach generalise to new unseen geometries?
* What are computatonal expenses/overhead in dependence on the important parameters?
* Why stds of your metrics are so high, especially for full rollout case? How can one increase the stability of the training?

---

### Note · Authors · 2025-12-03

I have read and agree with the venue's withdrawal policy on behalf of myself and my co-authors.